

# Detection of mild cognitive impairment based on attention mechanism and parallel dilated convolution

Tao Wang[1], Zenghui Ding[1], Xianjun Yang[1], Yanyan Chen[1], Yu Liu[2,3], Xiaoming Kong[2,3] and Yining Sun[1]

[1] Hefei Institutes of Physical Science, Chinese Academy of Sciences, Hefei, Anhui, China
[2] Affiliated Psychological Hospital of Anhui Medical University, Hefei, Anhui, China
[3] Hefei Fourth People's Hospital, Hefei, Anhui, China

## ABSTRACT

Mild cognitive impairment (MCI) is a precursor to neurodegenerative diseases such as Alzheimer's disease, and an early diagnosis and intervention can delay its progression. However, the brain MRI images of MCI patients have small changes and blurry shapes. At the same time, MRI contains a large amount of redundant information, which leads to the poor performance of current MCI detection methods based on deep learning. This article proposes an MCI detection method that integrates the attention mechanism and parallel dilated convolution. By introducing an attention mechanism, it highlights the relevant information of the lesion area in the image, suppresses irrelevant areas, eliminates redundant information in MRI images, and improves the ability to mine detailed information. Parallel dilated convolution is used to obtain a larger receptive field without downsampling, thereby enhancing the ability to acquire contextual information and improving the accuracy of small target classification while maintaining detailed information on large-scale feature maps. Experimental results on the public dataset ADNI show that the detection accuracy of the method on MCI reaches 81.63%, which is approximately 6.8% higher than the basic model. The method is expected to be used in clinical practice in the future to provide earlier intervention and treatment for MCI patients, thereby improving their quality of life.

# INTRODUCTION

As the trend of population aging continues to increase, cognitive impairment among the elderly population has attracted increasing attention. As a state between normal aging and Alzheimer's disease (AD), mild cognitive impairment (MCI) detection has important clinical and social significance. According to a report released by Alzheimer's Disease International, the number of people suffering from dementia worldwide in 2018 was approximately 50 million, and it is estimated that this will double by 2050, with one new person suffering from dementia globally every 3 s, two-thirds of the cases would be classified as AD (*Patterson, 2018*). The Centers for Disease Control (CDC) in the

Corresponding author
Zenghui Ding, dingzenghui@iim.ac.cn

United States believes AD has become the third leading cause of death after heart disease and cancer (*James et al., 2014*). According to a series of reports on economic and social development achievements for the 70th anniversary of the founding of the People's Republic of China released by China in 2019, the proportion of China's population aged 65 and over is as high as 11.9%, and the prevalence of AD is increasing year by year (*Ling et al., 2020*). It is expected that by 2050, the total number of AD patients in China will be close to 28 million, becoming one of the countries with the largest number of AD patients and the fastest-growing rate in the world (*Clay et al., 2019*). MCI is considered an early stage of AD. Compared with the general population, patients with MCI have a significantly higher probability of transforming to AD (*Tahami Monfared et al., 2022*). Early identification and intervention of MCI are expected to delay or prevent the development of AD (*Sabbagh et al., 2020*).

Currently, clinical MCI diagnosis usually relies on neuropsychological tests, such as Mini-Mental State Examination (MMSE), Montreal Cognitive Assessment (MoCA), Clinical Dementia Rating (CDR) and the patient's clinical manifestations are evaluated (*Shie et al., 2021*). This method is easy to operate and time-consuming, but its accuracy is easily affected by education level and involves a certain degree of subjectivity (*Molinuevo et al., 2017*). In recent years, advances in medical imaging technology have provided new opportunities for the diagnosis of MCI, such as electroencephalogram (EEG), diffusion tensor imaging (DTI), positron emission tomography (PET), magnetic resonance imaging (MRI), etc. (*Shukla, Tiwari & Tiwari, 2023*; *Subramanyam Rallabandi & Seetharaman, 2023*; *Yang et al., 2019*). Among them, MRI is a non-invasive medical imaging technology that is safe, harmless and has high resolution. It can display the anatomical and functional information of the brain in detail and is widely used in the diagnosis of neuropsychiatric diseases such as MCI and AD. Compared with the normal aging process, the brain structure of MCI patients will show abnormal enlargement and shrinkage, especially in specific areas such as the hippocampus and lower lateral ventricles. However, in clinical practice, it is often difficult to precisely locate these lesions by visual inspection. As a method that imitates the connection of neurons in the human brain, deep learning can automatically learn and extract features from data, and gradually understand complex patterns and relationships. This allows deep learning to capture subtle changes in complex neuroimaging data, revealing biological signatures associated with cognitive impairment. Therefore, the use of deep learning to learn and extract valuable features from large-scale medical imaging data to achieve automated detection of MCI detection has attracted widespread attention (*Fathi, Ahmadi & Dehnad, 2022*).

*Li et al. (2022)* proposed an MCI identification method based on a three-dimensional convolutional neural network (3-D CNN), which provides additional supervisory information for supervised classification tasks through multi-channel contrastive learning. *Odusami et al. (2021)* used fine-tuned ResNet18 to build a deep-learning network for MCI recognition. *Liu et al. (2020)* proposed an MCI classification method to enhance multi-modal MRI data feature representation by combining multi-view information. *Alyoubi et al. (2023)* used the entorhinal cortex area in MRI and combined different neural network architectures such as VGG16, Inception-V3 and ResNet50 to

build an MCI model. Although brain MRI images have the advantage of providing rich anatomical and functional information, there is a large amount of redundancy. During the feature extraction process, due to the complexity of the brain structure, significant lesion features are often obscured or ignored, resulting in the loss of useful information. To overcome this problem, *Qin et al. (2022)* proposed a DHA-ResUNet method combined with a hybrid attention mechanism to assist in the diagnosis of MCI. This method achieves better recognition and localization of features by fusing channel attention and spatial attention. *Chen, Qiao & Zhu (2022)* proposed an MCI diagnosis model based on multi-view slice attention and 3D-CNN, which emphasizes specific 2D slices through a slice-level attention mechanism to exclude redundant features. *Zhang et al. (2022)* proposed a deep learning framework based on sMRI gray matter slices for MCI diagnosis. This method uses the channel attention module to enhance the important information of the processed object and suppress some irrelevant details. Table 1 summarizes existing MCI detection methods.

Compared to natural images, medical images exhibit obvious target region localization characteristics. The lesion location typically occupies a relatively small area of the entire MRI image. Traditional methods integrate the channel attention mechanism and the spatial attention mechanism into the network in series. This type of method can achieve certain results in the depth of the network, but it ignores the influence of global features and often fails to perform well when processing fine-grained spatial information tasks. In this study, we propose a convolutional neural network method based on attention mechanism and parallel dilated convolution. Specifically, we introduce an attention mechanism to highlight relevant information of the lesion area and suppress irrelevant areas, thereby improving the ability to mine detailed information. At the same time, through parallel dilated convolution, we achieve a larger receptive field and enhance the ability to obtain contextual information. The main contributions of this study are summarized as follows:

1. Introducing an attention gate mechanism module suitable for medical images to eliminate redundant information in medical images, highlight diseased areas, and suppress the influence of irrelevant areas.
2. Use parallel dilated convolution to obtain a larger receptive field without downsampling, thereby enhancing the ability to obtain contextual information while maintaining detailed information on large-scale feature maps.
3. Experiments on the public data set ADNI proved that the accuracy of this method in MCI detection was significantly improved, providing a feasible solution for future clinical practice.

The rest of the article is organized as follows: the "Method" section describes our method. "Results" introduces the dataset, evaluation indicators and experimental settings, and demonstrates the effectiveness of the method through quantitative and qualitative analyses. "Discussion" discusses some key factors that influence the performance of the proposed method. Finally, "Conclusion" concludes our work.

**Table 1  Summary of MCI detection methods.**

| Author | Dataset | Subject | Methodology | Performance | Contribution |
|---|---|---|---|---|---|
| *Li et al. (2022)* | ADNI dataset (T1w MRI) | 928 subjects (330 NC, 299 AD, 299 MCI): 3746 MRI images | 3D CNN | • MCI *vs.* NC: ACC = 80.44%, SEN = 83.18%, SPE = 78.59%, PRE = 72.33%, AUC = 80.89%, F1 = 77.38%. | The study used a multi-channel contrastive learning strategy based on multiple data transformation methods (e.g., adding noise) to combine supervised classification loss with unsupervised contrastive loss to improve the classification accuracy and generalization ability of the network. |
| *Odusami et al. (2021)* | ADNI dataset (fMRI) | 138 subjects: 78,753 images | 2D ResNet18 | • CN *vs.* EMCI: ACC = 96.51%, SEN = 98.62%, SPE = 99.96%. <br> • CN *vs.* LMCI: ACC = 74.91%, SEN = 67.36%, SPE = 97.92%. | An improved ResNet18 model fine-tuning framework is proposed to achieve AD image classification of seven binary categories by extracting useful features in hippocampal fMRI data. |
| *Liu et al. (2020)* | ADNI dataset (T1w MRI+rs-fMRI) | 315 subjects (105 LMCI, 105 EMCI, 105 NC) | MTFS-gLASSO-TTR+multi-kernel learning | • LMCI *vs.* NC: ACC = 88.5%, SEN = 86.3%, SPE = 90.3%, AUC = 89.7%. <br> • EMCI *vs.* NC: ACC = 82.7%, SEN = 79.4%, SPE = 83.9%, AUC = 83.2%. | This study proposes a method to enhance feature representation of multi-modal MRI data to improve the performance of mild cognitive impairment (MCI) classification by combining multi-view information. |
| *Alyoubi et al. (2023)* | ADNI dataset (T1w MRI) | 188 subjects (95 NC, 93 MCI): 779 3D-MRI images | VGG16, Inception-V3 and ResNet50 | • The best model's (Inception-V3) MCI *vs.* CN: ACC = 70%, F1 = 0.73%, SEN = 90%, SPE = 54%, AUC = 69%. | The study noted that the parahippocampal cortex already showed underlying changes before hippocampal atrophy. This means that the parahippocampal cortex can serve as a key area for early diagnosis of mild cognitive impairment (MCI), providing the possibility for early intervention and treatment. |
| *Qin et al. (2022)* | ADNI dataset | 43 aMCI, 46 sMCI, 5 oMCI | 3D HA-ResUNet | •aMCI *vs.* sMC: ACC = 100%, SEN = 100%, SPE = 100%, PRE = 100%, F1 = 100%, G-mean = 100%. | This study adopted an attribution-based visual interpretability method to reveal the regions and features used by the model for classification, providing a valuable reference for physicians' clinical decision-making. |
| *Chen, Qiao & Zhu (2022)* | ADNI-1 (1.5 T T1W sMR) and ADNI-2 (3T T1W sMR) datasets | • ADNI-1: 808 subjects (183 AD, 229 CN, 167 pMCI, 229 sMCI). <br> • ADNI-2: 643 subjects (143 AD, 184 CN, 75 pMCI, 241 sMCI). | Multiview-Slice Attention and 3D CNN | • pMCI *vs.* sMCI: ACC = 80.1%, SEN = 52.0%, SPE = 85.6%, AUC = 78.9%. | This study considers the characteristics of multi-view slices and feature complementarity, and proposes a method to comprehensively utilize multi-dimensional slice features, allowing the model to extract information more effectively from magnetic resonance imaging. |
| *Zhang et al. (2022)* | ADNI dataset (sMRI) | 496 subjects (139 AD, 198 MCI, 159 NC) | Attention mechanism + 2D CNN | • MCI *vs.* NC: ACC = 67.1%, SEN = 80.0%, SPE = 53.1%. | This method enhances gray matter feature information through the combination of slice areas and attention mechanisms, thereby improving the accuracy of AD diagnosis. |

# METHOD

The development of deep neural networks has led to a huge leap in the field of artificial intelligence. However, as the number of network layers increases, the vanishing gradient problem gradually appears. This problem makes it difficult to update the

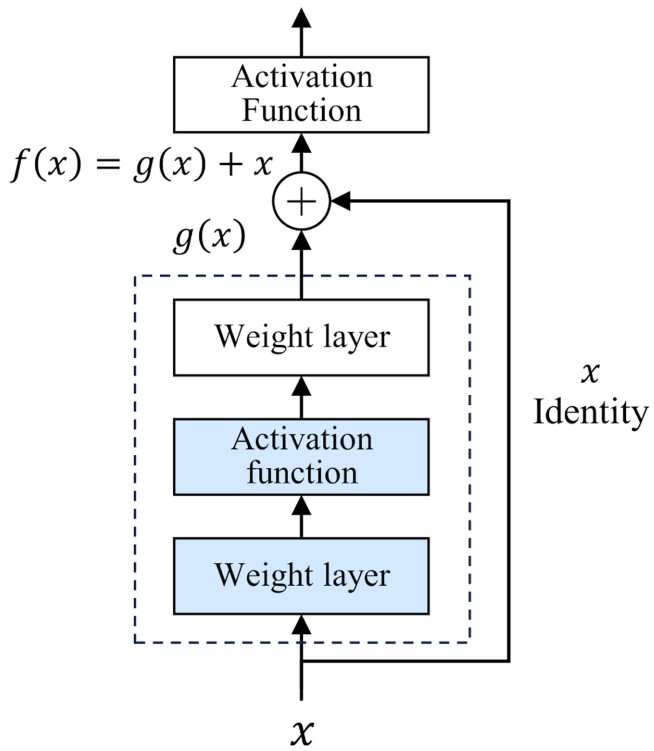

**Figure 1  Residual structure.**

underlying network parameters, thus limiting further improvements in network depth and performance. To deal with this challenge, *He et al. (2016)* proposed a deep residual network (ResNet) in 2015. The core idea of ResNet is "residual learning". Unlike traditional neural networks, which directly fit the input and output mapping layer by layer, ResNet learns the residual between the output of the previous layer and the input of the current layer, and adds the residual to the output of the previous layer to realize the transmission and capture of information. The structure of ResNet is shown in Fig. 1.

This design has two-fold advantages. First, the residual module constructs a shortcut for information dissemination. In traditional networks, information must pass through a series of intermediate-weight layers. As the number of network layers increases, the propagation loss of shallow features gradually accumulates. However, in ResNet, the introduction of residuals greatly reduces the depth of information propagation in the network, effectively reducing the loss of shallow features. Secondly, the iteration of the residual module strengthens the ability to abstract features, which makes the network more capable of deeply characterizing and abstracting the input data. In 2015, ResNet won the championship in the ImageNet competition, reducing the Top-5 error rate to 3.57%. Classic ResNet ranges from 18 to 152 layers (*He et al., 2016*), and its performance has been verified at different scales. This study uses the basic ResNet18 as the backbone framework.

## Attention gate mechanism

The attention mechanism is a technology that simulates the operation of the human brain. Its purpose is to imitate the selective attention characteristic phenomenon that humans present in the process of information processing. This mechanism allows humans to selectively focus on specific sources of information and thereby ignore information that is irrelevant to the current task. Using this ability, humans can gain a more refined and in-depth understanding of complex situations. With the continuous progress in the fields of computer science and artificial intelligence, researchers have introduced attention mechanisms into the field of deep learning to optimize the execution performance of various tasks. Among them, the most representative works include the channel attention mechanism proposed by *Hu, Shen & Sun (2018)* in 2017, the CBAM mechanism that integrates channel attention and spatial attention proposed by *Woo et al. (2018)* in 2018, and the 2020 (*Wang et al., 2020*) improved the channel attention mechanism, *etc.* By generating weighted attention maps in the two dimensions of channel and space, the network can more effectively focus on key channel features and spatial location information. This optimization method has achieved remarkable results in the field of natural image processing.

However, compared with natural images, medical images have the characteristic of localized target areas. Especially for the brain MRI images used in this study, the location of the lesions only occupies a small area of the entire brain MRI image. Currently, methods using channel attention mechanisms can automatically learn the importance between different channels (feature maps), allowing the network to focus on information useful for solving specific tasks. However, it often performs unsatisfactorily when processing tasks with fine-grained spatial information. While methods based on spatial attention mechanisms ignore the influence of global features. In the diagnosis of MCI, brain MRI lesions show obvious localization characteristics, and early lesions are not obvious. Therefore, during the diagnosis process, local information and global information need to be effectively combined.

To address this problem, this study introduces an attention mechanism method that targets the localization characteristics of targets in medical images, that is, the attention gate mechanism (AG) (*Schlemper et al., 2019*). The structure of the attention gate mechanism is shown in Fig. 2. Among them, $x_l$ is the local feature vector extracted from the middle layer of the network, and $g_l$ represents the global feature vector obtained from the coarse-scale part of the network, which contains abstract information of the target, such as size, position and orientation. The global feature $g_l$ is used as a gating signal, and the local feature $x_l$ is sent to the gated attention. The attention coefficient $\alpha^l$ is calculated, and its value is between the interval $[0, 1]$. The calculation formula of attention coefficient $\alpha^l$ is as follows:

$$q_{att,i}^l = \Psi^T \left( \sigma_1 \left( W_x^T x_i^l + W_g^T g + b_{xg} \right) \right) + b_\Psi$$
$$\alpha^l = \sigma_2 \left( q_{att}^l \left( x^l, g; \Theta_{att} \right) \right) \tag{1}$$

where $\sigma_1$ and $\sigma_2$ represent different activation functions. $\sigma_1$ is a rectified linear unit (ReLU), and $\sigma_2$ is generally a Softmax function to ensure that the sum of attention coefficient

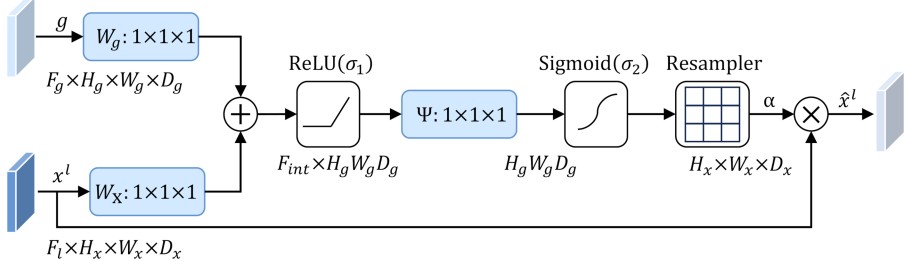

**Figure 2  Attention gate mechanism.**

$\alpha_i^l = e^{q_{att,i}^{q_i^l}} / \sum_i e_{att,i}^{q^l}$ is 1. The output of the attention gate mechanism is completed by a channel-based $1 \times 1 \times 1$ convolutional linear transformation, including three parts: $W_x \in \mathbb{R}^{F_l \times F_{int}}$, $W_g \in \mathbb{R}^{F_g \times F_{int}}$ and $\Psi \in \mathbb{R}^{F_int \times l}$. The corresponding bias terms are $b_{xg} \in \mathbb{R}^{F_{int}}$ and $b_\Psi \in \mathbb{R}$, respectively. The final weighted output is $\hat{x}^l = \{\alpha_i^l x_i^l\}_{i=1}^n$. It can be seen that in the target area, the attention coefficient is larger, while in the background area, the attention coefficient is smaller. Therefore, the method is able to extract feature responses relevant to the target task from medical images and suppress the influence of useless features.

## Parallel dilated convolution

An important challenge facing the field of MCI detection is that the lesion area appears localized and its size is usually extremely small. Therefore, ensuring that these small-size target features are not ignored during the recognition process has become one of the key factors to ensure accurate recognition of the model. Classic classification networks at this stage usually add pooling layers before and after the convolutional layer to increase the receptive field of the network. However, the pooling operation will lead to the loss of some spatial information, including important detailed features that may be included in the original image, and may even directly ignore small-scale lesion features.

Dilated convolution, also known as atrous convolution, was first proposed by *Yu & Koltun (2015)* in 2016. It was originally used to solve the problem of pooling operation reducing image resolution and causing the loss of some features in image semantic segmentation, as an alternative to the pooling layer. In a convolutional neural network, if you directly remove the pooling layer that may reduce the image size, you will not be able to increase the receptive field of the small-sized convolution kernel, and if you blindly increase the size of the convolution kernel, it will significantly increase the number of parameters of the network. By using a sparse kernel, dilated convolution expands the effective size of the convolution kernel without increasing network parameters, thereby expanding the receptive field of the network. This effect is similar to alternating convolutional and pooling layers, but does not reduce the size of the feature map.

The basic principle of dilated convolution is to expand the convolution kernel parameters that are originally closely connected according to a certain ratio, and use zero values to separate these parameters. The separation distance is determined by the expansion rate. The advantage of this design is that the receptive field of the network is expanded through the

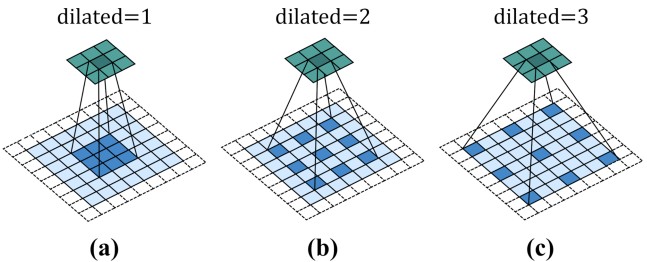

dilated=1       dilated=2       dilated=3

(a)           (b)           (c)

**Figure 3**   **Dilated convolution.**

expansion of the convolution kernel, but the kernel parameters involved in the calculation do not increase, thus not increasing the computational burden. At the same time, the size of the feature map after dilation convolution remains unchanged, effectively retaining important detailed features.

Figure 3 shows a $3 \times 3$ dilated convolution kernel with different dilation rates. It can be observed from the figure that when the dilation rate is 1, dilated convolution is no different from ordinary convolution. When the dilation rate is 2, the interval between convolution kernel elements is 1. After filling, the dilated convolution kernel is equivalent to a $5 \times 5$ convolution kernel. When dilated $= 3$, the interval between convolution kernel elements is 2, which is equivalent to a receptive field of $7 \times 7$ convolution kernel. That is to say, for a convolution kernel with an original size of $k \times k$, when the expansion rate is $d$, the actual side length of the convolution kernel will become $k + (k - 1) \times (d - 1)$.

For a two-dimensional input data, the formula of dilated convolution is defined as follows:

$$y(m, n) = \sum_{i=1}^{M} \sum_{j=1}^{N} x(m + r \cdot i, n + r \cdot j) w(i, j) \tag{2}$$

where $x(m, n)$ represents the input of the network, $w(i, j)$ represents the filter of size $M \times N$, and $r$ represents the expansion rate of the convolution kernel, and $y(m, n)$ is the output of the network. By introducing dilated convolution, a larger receptive field can be obtained without the need for downsampling. This enables the retention of detailed information and the rich acquisition of contextual information on large-scale feature maps.

To solve the problem of small-size target feature recognition in MCI detection, this study introduces a parallel dilated convolution module, that is, Atrous Spatial Pyramid Pooling (ASPP), into the network structure. The parallel dilated convolution module contains multiple dilated convolution operations, and each operation uses a different dilation rate. The output feature map of each dilated convolution operation preserves different scale contextual information. By using the feature fusion method, the output feature maps of each dilated convolution operation are stacked in the channel dimension to obtain richer multi-scale feature representations. The ASPP is shown in Fig. 4.

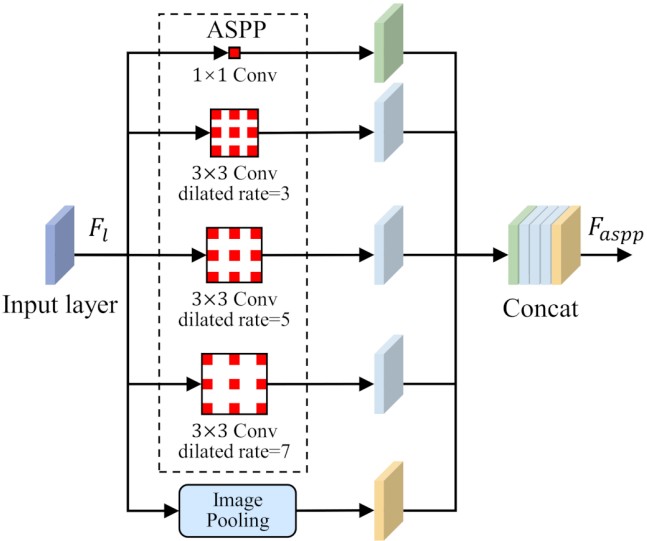

**Figure 4**   Atrous Spatial Pyramid Pooling (ASPP).

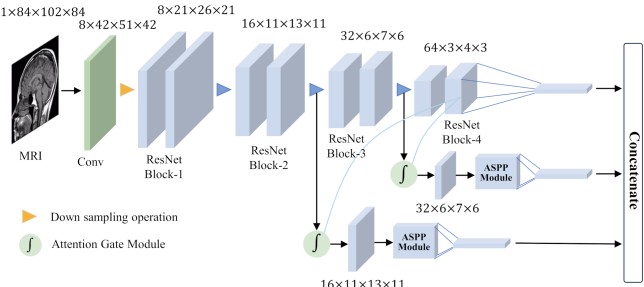

**Figure 5**   Overall network architecture.

## Overall network architecture

To achieve good classification results for small target lesions, the network needs to obtain rich lesion area information, including the shape and size of the lesion and other features. These features usually exist in shallow networks, but because the lesion area is too small, these shallow features will lose a lot of detailed information during the feature extraction process. In this study, based on brain MRI images, we proposed an improved ResNet18 network for the detection of MCI with the help of an attention gate mechanism and parallel dilated convolution. As shown in Fig. 5, this network utilizes the deepest feature map $g$ as a gating signal to provide contextual information for the feature maps of the second and third ResNet blocks. The purpose of this operation is to prune redundant information in shallow features to highlight the salient features of the lesion area. The deepest feature map $g$ is selected as the gating signal to ensure that the extracted contextual information

is highly correlated with the deep features, thereby helping to accurately identify the lesion area.

After eliminating redundant features, these features are input to the ASPP, and the detailed information and contextual information of the large-scale feature map are extracted through dilation convolution with different dilation rates. This step is crucial as it allows the network to focus on features at multiple scales simultaneously, thus better capturing the diversity of the lesion area. After obtaining the output of the ASPP, it is fused with the gating signal $g$ to further obtain the features of fused gating information and multi-scale information. This fused feature has a larger scale range, which helps avoid the loss of detailed features and effectively solves the problem of poor performance of existing methods in classifying small target lesions.

# RESULTS

## Dataset
### ADNI dataset
The data used in this study are derived from the Alzheimer's Disease Neuroimaging Initiative (ADNI) database (*Mueller et al., 2005*), which was created in 2004 and supported by the National Aging National Institute of Biotechnology (NIA), National Institute of Biomedical Imaging and Bioengineering (NIBIB) and several pharmaceutical companies and institutions with financial support. ADNI's primary research focus is tracking the progression of early-stage AD and MCI. The database contains information on subjects' MRI, PET, genetics, cognitive tests, cerebrospinal fluid and blood biomarkers. In this study, we selected 3.0T MRI images of subjects in the ADNI database. These MRI images were acquired with a scanner manufactured by SIEMENS, using the 3D MPRAGE protocol to obtain sagittal MRI images. The images have an in-plane spatial resolution of $1.0 \times 1.0 \text{ mm}^2$ and a sagittal slice thickness of 1.2 mm. Considering that the subjects' longitudinal examination data also contains valid information. Therefore, we performed longitudinal time point acquisition and obtained 265 MRI samples from 127 MCI patients and 265 MRI samples from 75 normal controls (NC). Demographic information of subjects in the ADNI database is presented in Table 2. Due to the large inter-subject variability of MRI, to make the data distribution of the training set and the test set as close as possible while avoiding data leakage, we took 25% of the subjects as the test set and the remaining 75% as the training set.

### Image preprocessing and data augmentation
Due to the high dimensionality of MRI and relatively sparse medical data, deep learning algorithms face huge challenges in their training and convergence processes. Therefore, all images must be preprocessed to map brain image samples to a common coordinate space. This study uses the CAT12 toolkit of SPM12 (*Gaser et al., 2022*) to perform the necessary processing on the above MRI images: (1) Skull stripping. Use the built-in skull stripping function of CAT12 software to remove skull structures to exclude brain skull information that is not relevant to the experiment; (2) Tissue segmentation. Segment the MRI image into three tissue types: white matter, gray matter and cerebrospinal fluid.

**Table 2 Demographic information of subjects.**

| Diagnosis | Subjects | Gender (M /F) | Age (Mean ± SD) | Scans |
|---|---|---|---|---|
| NC | 127 | 64/63 | 75.72 ± 6.87 | 265 |
| MCI | 75 | 46/29 | 77.99 ± 8.10 | 265 |

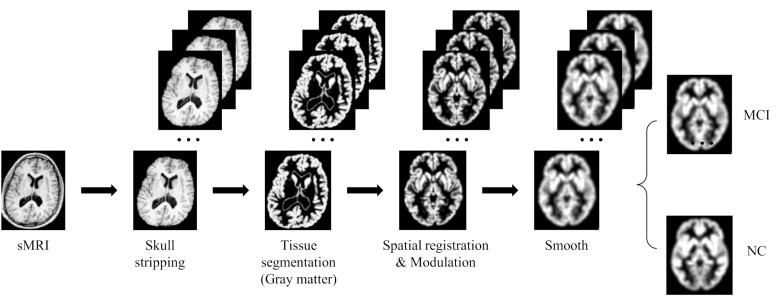

**Figure 6 MRI image preprocessing process.** Figure source credit: ADNI.

Since the gray matter tissue of the brain is susceptible to morphological changes caused by MCI, this study uses the gray matter tissue of the subject's brain as input; (3) Spatial registration. Register the subject's image to the MNI152 standard template established by the Montreal Neurological Institute-Hospital (MNI) to reduce the differences in individual brain spaces; (4) Modulation. Ensure that brain tissue is comparable in template space while retaining differences between individuals; (5) Smooth. Spatial smooth is performed on the image to reduce reconstruction errors and improve the consistency of the subject's brain image. After the above preprocessing steps, a brain MRI gray matter image with a size of $84 \times 102 \times 84$ was finally obtained. Figure 6 gives a schematic diagram of the preprocessing process.

Deep learning methods are known for their powerful potential and highly complex models, however, for these models to produce robust results, large image data sets are usually required. Due to the high cost and difficulty of obtaining clinical data, enough data cannot be obtained. In this study, we use data augmentation techniques such as flipping, rotating, and adding noise to improve the training effect of the neural network model. Among them, flipping the image vertically, horizontally and axial can increase the mirrored version of the data, helping the model better capture different perspectives and features in the image. The rotation operation can introduce more changes, making the model better able to recognize objects at different angles. By adding random noise to the image to simulate the uncertainty in the real world, it helps to improve the robustness of the model and make it better able to deal with noise and interference in the real world. In this study, we utilize these data augmentation techniques to expand the training data by 25 times. This significantly improves the model's generalization ability on unknown data.

## Experimental setup

The experimental running system of this study is Ubuntu 20.04.6 LTS, the CPU is 12th Gen Intel (R) Core (TM) i9-12900K with a main frequency of 3.9 GHz, the memory is

64 GB, the GPU model is NVIDIA GeForce RTX 3080*2, and the video memory is 24 GB. The experimental environment is python3.8 and pytorch1.7.1. In the experiment, the three-dimensional convolution kernel and fully connected weight parameters were initialized with truncated normal distributed random numbers with a standard deviation of 0.1. For the setting of hyperparameters, based on our experience and previous research, we set the batch size to 8. Compared to larger batch sizes, smaller batch sizes promote model convergence more effectively and are more efficient in terms of memory utilization. For the choice of optimizer, we use the Adam optimizer. The Adam optimizer combines the ideas of momentum and RMSprop, has the characteristics of adaptive learning rate, and has outstanding performance in handling many tasks. At the same time, compared to other optimizers, Adam is less sensitive to the selection requirements of hyperparameters. For the learning rate, we follow the default value used in most current studies, which is 0.001. In addition, we also adopt a learning rate decay strategy, that is, after every 10 epochs, the learning rate is adjusted to 0.1 times its original value. This strategy can improve the performance and stability of the optimization algorithm, allowing the model to converge to a better solution faster. The total training epochs are set to 50.

## Evaluation indicators

This study uses common evaluation indicators in medical image classification tasks to evaluate the performance of the model, including sensitivity (SEN), specificity (SPE), accuracy (ACC) and F1-score (F1). The definitions of each indicator are as follows:

$$SEN = \frac{TP}{TP+FN} \tag{3}$$

$$SPE = \frac{TN}{TN+FP} \tag{4}$$

$$ACC = \frac{TP+TN}{TP+TN+FP+FN} \tag{5}$$

$$f1 = \frac{2 \times \frac{TP}{TP+FP} \times SEN}{\frac{TP}{TP+FP} + SEN}. \tag{6}$$

Among them, TP, TN, FP and FN represent true positive, true negative, false positive and false negative, respectively. In addition, the area under the curve (AUC) is also introduced to evaluate the overall classification performance of the model. AUC is the area enclosed by the receiver operating characteristic (ROC) curve in the $[0, 1]$ interval and the $X$-axis, which reflects the overall performance of the model. The larger the AUC value, the better the model classification performance.

## Experiment analysis

Figures 7 and 8 depict the training-validation loss and accuracy of the proposed method across iteration epochs, along with the confusion matrix and ROC curve. First, from Fig.
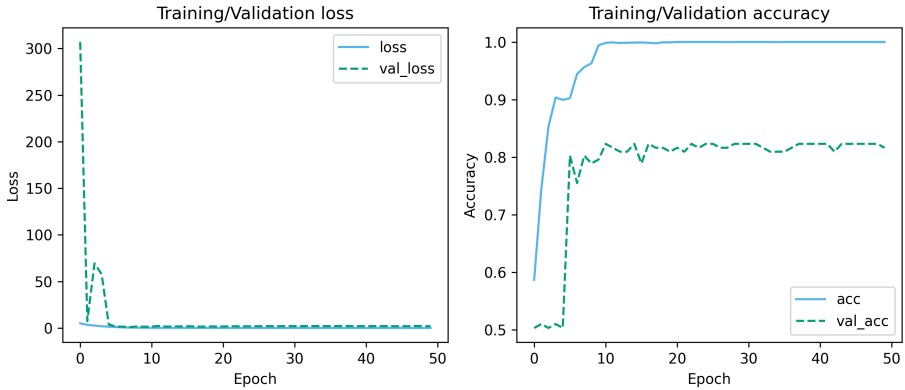

**Figure 7  Training-validation loss and accuracy over iteration epochs.**

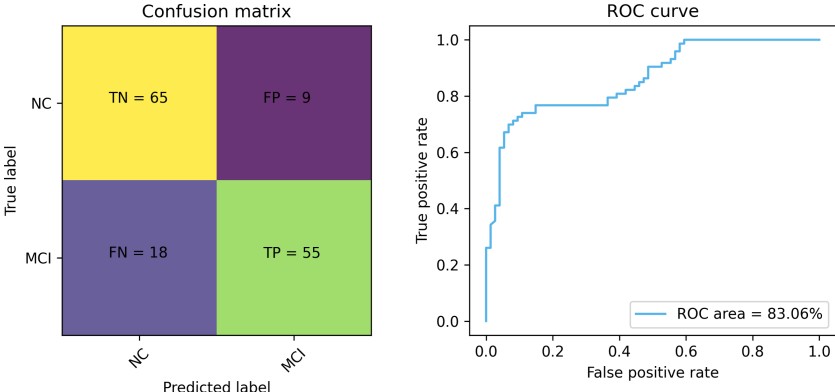

**Figure 8  The confusion matrix and ROC curve of the proposed method.**

7, it can be found that as the number of iterations increases, the training and validation losses of the model gradually decrease and the accuracy gradually increases. In particular, the proposed method reaches stable accuracy after about 15 epochs. This shows that the proposed model can effectively reduce errors during the learning process and reach acceptable performance in a relatively short time. However, we also noticed a large difference in accuracy between the training and validation sets. This difference can be attributed to the smaller sample size used in the study. As the number of iterations increases, the model overfits on the training set. Further observing the confusion matrix and ROC curve in Fig. 8, we can find that out of 147 test samples, 120 were successfully predicted. This indicates that the proposed method achieves good performance in detecting MCI. In addition, we also showed the ROC curve with an AUC value of 83.06%, which further proved its effectiveness in detecting MCI.

## Comparison of different ResNet architectures

To verify the performance differences on different ResNet architectures, we compared several typical ResNet basic frameworks, including ResNet18, ResNet34, ResNet50,

**Table 3   The performance comparison between different ResNet architectures.**

| Architecture | SEN | SPE | ACC | AUC | F1 |
|---|---|---|---|---|---|
| ResNet18 | 63.01 | 86.49 | 74.83 | 83.08 | 71.32 |
| ResNet34 | 53.42 | 94.59 | 74.15 | 81.90 | 67.24 |
| ResNet50 | 64.38 | 83.78 | 74.15 | 80.64 | 71.21 |
| ResNet101 | 61.64 | 77.03 | 69.39 | 79.67 | 66.67 |
| ResNet152 | 58.90 | 82.43 | 70.75 | 76.03 | 70.75 |

ResNet101, and ResNet152. These different ResNet architectures have obvious differences in depth, parameter volume and computational complexity, and have different impacts on MCI detection tasks. The experimental results are shown in Table 3. We used a variety of performance evaluation indicators to comprehensively evaluate the performance of different ResNet architectures, including sensitivity, specificity, accuracy, F1 and AUC.

From Table 3, it can be observed that ResNet18, ResNet34 and ResNet50 perform equally well in terms of accuracy. However, ResNet101 and ResNet152, two relatively deep ResNet architectures, show a clear downward trend in accuracy. This phenomenon is because as the depth of the model increases, the network overfits, resulting in a decrease in generalization performance on the test data. In addition, comparing ResNet18 and ResNet50, it can be found that they are generally equivalent in various indicators. However, compared with ResNet50, ResNet18 has a significantly lower number of parameters and computational complexity, and has more advantages in model lightweight, which is of great significance for MCI detection tasks in resource-constrained environments. Therefore, in this study, we choose ResNet18 as the backbone framework for the task.

## Comparison of different attention modules

To verify the effectiveness of the attention module used in this study, we embedded different types of attention modules into the ResNet-18 backbone framework and compared their impact on model classification performance. In these experiments, we include a variety of widely used attention modules, such as SENet (*Hu, Shen & Sun, 2018*), ECANet (*Wang et al., 2020*) and CBAM (*Woo et al., 2018*), to ensure comprehensive evaluation, and the experimental results are shown in Table 4.

Combining the results in Tables 3 and 4, we can clearly observe that after integrating different types of attention modules into the network, the overall performance has improved. This performance improvement is reflected in multiple key performance indicators, including sensitivity, accuracy and F1. However, it is worth noting that compared to other attention modules, the attention gate mechanism (AG) adopted in this study shows compelling superiority in terms of accuracy and F1. This is because the attention gate mechanism is a method proposed for the localization characteristics of the target area of medical images. The method enables the network to focus more on the area of interest, extract feature responses that are closely related to the task target, and effectively suppress the influence of useless features, thereby better capturing key features in medical images and improving the overall network performance.

**Table 4    The performance comparison between different attention modules.**

| Attention module | SEN | SPE | ACC | AUC | F1 |
|---|---|---|---|---|---|
| SENet | 65.75 | 85.14 | 75.51 | 81.58 | 72.73 |
| ECANet | 67.12 | 85.14 | 76.19 | 81.84 | 73.68 |
| CBAM | 72.60 | 79.73 | 76.19 | 82.32 | 75.18 |
| AG | 72.60 | 81.08 | 76.87 | 81.77 | 75.71 |

**Table 5    The impact of different modules on the overall model performance.**

| Method | SEN | SPE | ACC | AUC | F1 |
|---|---|---|---|---|---|
| Baseline | 63.01 | 86.49 | 74.83 | **83.08** | 71.32 |
| Baseline+AG | 72.60 (**+9.59**) | 81.08 (−5.41) | 76.87 (**+2.04**) | 81.77 (−1.31) | 75.71 (**+4.41**) |
| Baseline +AG+ASPP | **75.34 (+12.33)** | **87.84 (+1.35)** | **81.63 (+6.8)** | 83.06 (−0.02) | **80.29 (+8.97)** |

**Notes.**
The bold text represents the performance improvement over the baseline model.

## Ablation experiment

To prove the effectiveness of the module design in the model proposed in this study, we conducted an ablation experiment. The control variable method is used to gradually add different modules of the model and evaluate the impact of each module on the overall model performance. The experiment uses the ResNet18 basic network as the evaluation baseline. The results are shown in Table 5.

It can be clearly observed from Table 5 that with the gradual increase of different modules, the overall performance shows a gradual improvement trend. These results indicate that each module has a positive impact on the performance of the model. It is particularly worth noting that after adding the ASPP, the accuracy rate increased by about 4.76%. This proves that this module can better capture important information in images by introducing receptive fields of different scales, thereby improving classification performance.

## Comparison with other methods

To further verify the effectiveness of the method, we compared it with other work in the literature. It is worth noting that the ADNI database contains longitudinal examination data. To avoid data leakage leading to model overfitting, in this study we divided the data based on subjects. In the comparison, we also only report methods that explicitly adopt this strategy in the literature. Table 6 shows the performance comparison between the method and existing work.

As can be seen from Table 6, the method used in this study has achieved significant improvements in specificity and accuracy. Specificity and accuracy are key indicators for measuring the performance of a classification model. High specificity means that the model is more able to correctly classify healthy samples as healthy, while high accuracy means that the model is more able to correctly classify diseased samples as diseased. These results indicate that our method is more accurate in distinguishing normal samples and MCI patients, which is of great significance for early diagnosis and treatment of MCI.

**Table 6   The performance comparison between the method and existing work.**

| Study | Subject | | Classifier | SEN | SPE | ACC | AUC | F1 |
|---|---|---|---|---|---|---|---|---|
| | MCI | NC | | | | | | |
| *Zhang et al. (2022)* | 198 | 159 | ResNet | 80.00 | 53.10 | 67.10 | – | – |
| *Marzban et al. (2020)* | 106 | 185 | 2D CNN | – | – | 79.60 | 84.00 | – |
| *Heising & Angelopoulos (2022)* | 212 | 165 | 2D CNN | 89.40 | – | 73.50 | – | 80.20 |
| *Zhang et al. (2023)* | 768 | 459 | MRN | 89.13 | 50.41 | 73.77 | 73.14 | 80.39 |
| *Aderghal et al. (2020)* | 672 | 627 | LeNet | 77.72 | 81.44 | 78.48 | – | – |
| Our | 75 | 127 | ResNet18-AG-ASPP | 75.34 | 87.84 | 81.63 | 83.06 | 80.29 |

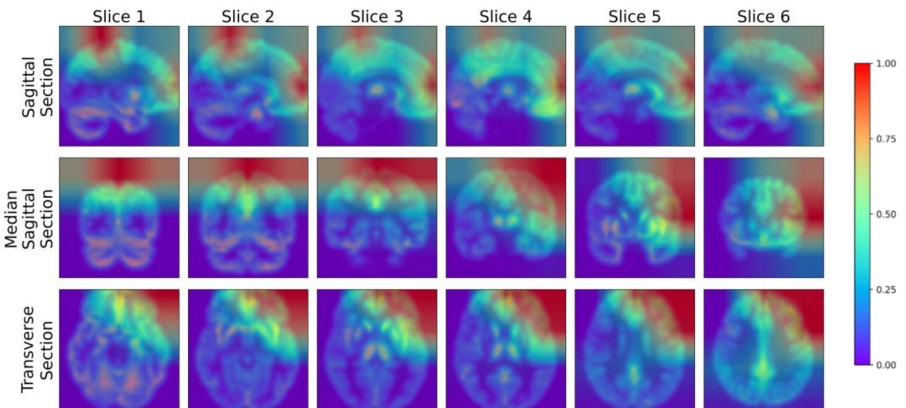

**Figure 9   Grad-CAM visualized lesion area.** Figure source credit: ADNI.

## Visualization

To further analyze the performance of the model proposed in this study, we adopted the Gradient-weighted Class Activation Mapping (Grad-CAM) technology (*Selvaraju et al., 2017*) to visualize the model's performance in key areas of interest, thereby qualitatively analyzing the effectiveness of the model in learning the characteristics of the lesion area. The Grad-CAM generated by this study's model is presented in Fig. 9, where blue represents low-weighted regions and red represents high-weighted regions. It can be observed from Fig. 9 that subjects have significantly high weights in the frontal and parietal regions. These brain regions are closely related to key functions in cognitive processes, including decision-making, planning, attention, working memory, and language comprehension. The early stages of MCI are often accompanied by atrophy of these brain areas. To verify the findings of the model, we further cooperated with professional doctors to carefully confirm the brain lesions of the subjects. These results strongly demonstrate that the model proposed in this study performs well in learning and utilizing lesion area information. It can extract effective features from these key regions, enabling highly accurate classification of MCI.

**Table 7   The performance under different backbone networks.**

| Backbone network | SEN | SPE | ACC | AUC | F1 |
|---|---|---|---|---|---|
| LeNet5 | 73.97 | 77.03 | 75.51 | 84.15 | 75.00 |
| AlexNet | 63.01 | 86.49 | 74.83 | 85.56 | 71.32 |
| GoogLeNet | 76.71 | 75.68 | 76.19 | 85.30 | 76.19 |
| VGG16 | 65.75 | 89.19 | 77.55 | 88.04 | 74.42 |
| ResNet18 | 75.34 | 87.84 | 81.63 | 83.06 | 80.29 |

## DISCUSSION

In the previous sections, we verified the effectiveness of the proposed method. In this section, we further discuss the key factors that affect the performance of the proposed method, including the backbone network, data augmentation, hyperparameters, and dataset distribution.

### Impact of backbone network on performance

Besides ResNet, deep learning networks such as LeNet, AlexNet, GoogLeNet and VGG are also widely used as backbone networks. To gain a deeper understanding of the impact of different backbone networks on model performance, we conducted a comparative analysis of LeNet5, AlexNet, GoogLeNet, VGG16 and ResNet18. The experimental results are shown in Table 7. First, as a classic deep learning model, LeNet5 shows relatively high sensitivity, but is slightly insufficient in other indicators. AlexNet performs well in terms of specificity and AUC, but its sensitivity and accuracy are slightly insufficient. GoogLeNet shows balanced performance, especially in AUC and F1 indicators. VGG16 achieved the best performance in specificity and AUC, showing its adaptability to MCI detection tasks. Finally, ResNet18 achieved excellent performance in accuracy and F1, demonstrating its advantages in handling complex tasks. It can be seen that choosing an appropriate backbone network is crucial to the performance of the model. One of the possible reasons why VGG16 performs well in this task is that its deep structure is better able to extract and characterize the features of the data. ResNet18, through its deep residual connection design, shows better stability and robustness when handling more complex tasks.

### Impact of data augmentation on performance

Data augmentation plays a vital role in the field of deep learning. Especially when clinical data sets are small, data augmentation can not only improve classification accuracy but also reduce the risk of overfitting. To verify the effectiveness of the data augmentation operation used in this study, we show the classification results of our method before and after data augmentation in Fig. 10. As can be seen from the figure, after using data augmentation operations, the classification results have improved in various indicators. First, without data augmentation, the classification accuracy is 77.55%, and with data augmentation, the classification accuracy increases to 81.63%. This shows that data augmentation can significantly improve the accuracy, thereby improving the reliability of clinical data analysis. In addition, in terms of sensitivity and specificity, the data augmentation operation has

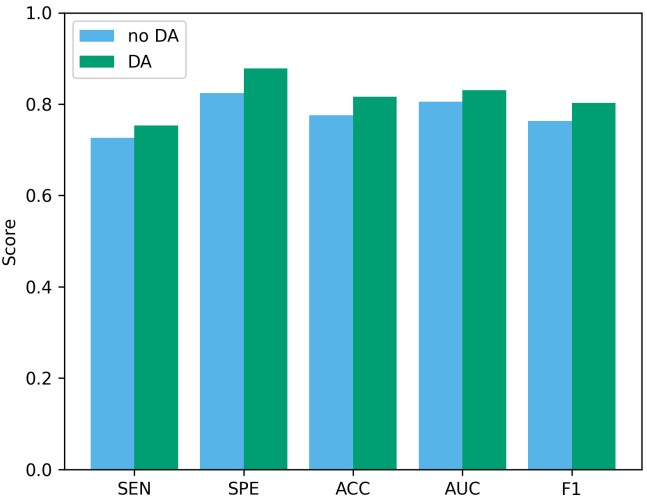

**Figure 10  The performance before and after data augmentation.** no DA, No data augmentation; DA, Data augmentation.

also been improved, which further confirms the effect of data augmentation in improving model performance.

## Impact of hyperparameters on performance

Hyperparameters play a vital role in deep learning, and they directly affect the performance and training process of the model. In this part, we analyze the impact of two hyperparameters, optimizer and batch size, on performance.

The batch size determines the number of samples used at each step in the training process. We explored the performance when batch sizes were 8, 16, 32, 64, 128, and 256. Figure 11 shows the performance of two key indicators (ACC and F1) at different batch sizes. As can be seen from Fig. 11, as the batch size increases, the accuracy and F1 decrease slightly. This is caused by the different gradient update frequencies under different batch sizes. Larger batch sizes usually mean fewer gradient updates, which can cause the model to converge slower during training, thus affecting performance indicators. In this study, the model performed best when the batch size was 8. Although the ACC are slightly lower at batch size 8 relative to some other batch sizes, the F1 performs best. This is because when the batch size is 8, the model can more fully capture the subtle features of the data and update the weights more frequently during the training process, resulting in better performance.

Besides batch size, the optimizer is another hyperparameter that often receives attention. In this study, we compare four common optimizers: stochastic gradient descent (SGD), root mean square propagation (RMSprop), Momentum, and Adam, to evaluate their impact on model performance. The experimental results are shown in Fig. 12. It can be seen that from Fig. 12, SGD showed good results in terms of sensitivity, accuracy and F1, but was slightly insufficient in specificity and AUC. RMSprop showed high specificity but relatively low sensitivity, accuracy and F1. The momentum optimizer performs better

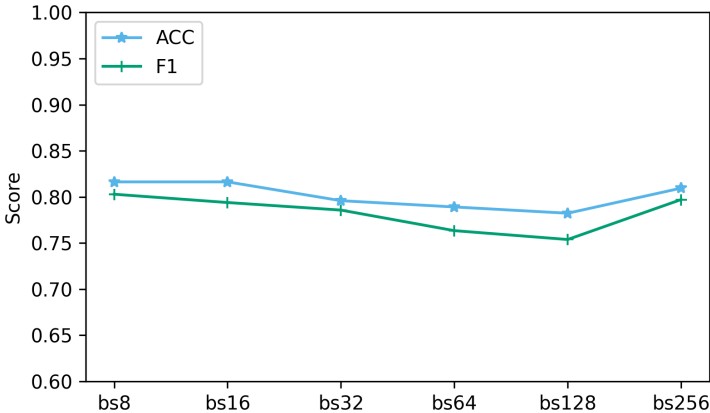

**Figure 11 The effect of different batch size on performance.** "bs" represents the batch size, and bs8 means the batch size equal to 8.

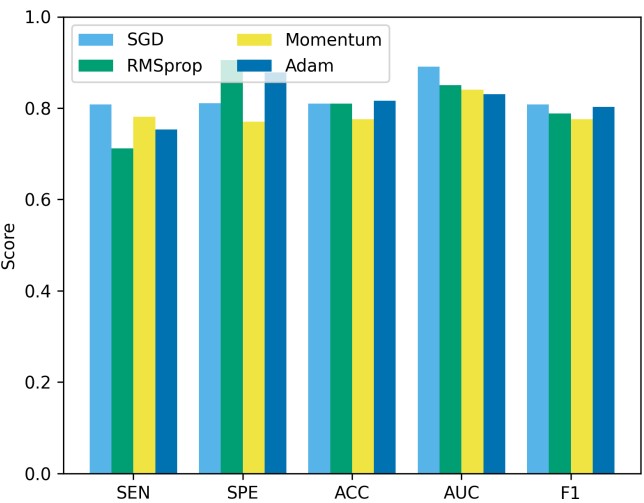

**Figure 12 The performance comparison using different optimizers.**

in terms of sensitivity and F1, but slightly lower specificity and accuracy. The Adam optimizer performs best in terms of accuracy and AUC, and is also competitive in other indicators. The Adam optimizer combines the ideas of momentum and RMSprop, has the characteristics of adaptive learning rate, and introduces a momentum term to accelerate convergence. This makes Adam perform well in many tasks, especially when dealing with complex non-convex optimization problems. Therefore, we choose to use the Adam optimizer in this study, and compared with other optimizers, it is less sensitive to the selection requirements of hyperparameters.

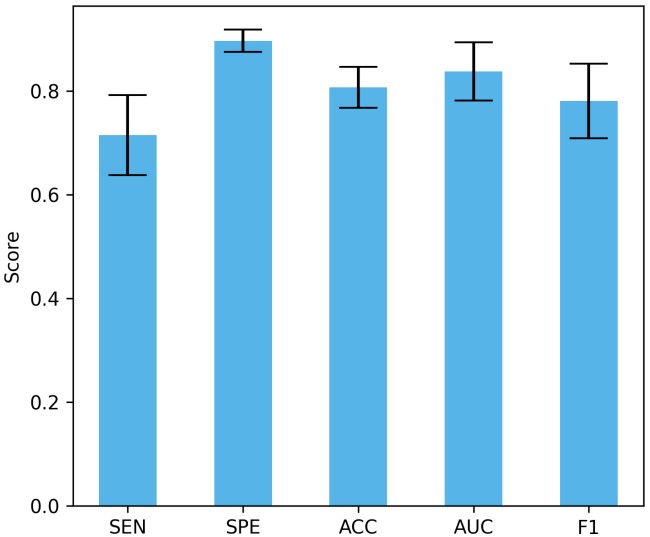

**Figure 13** **The performance under different dataset distributions.**

## Robustness under different dataset distributions

Differences in data distribution may lead to changes in model performance, so verifying the performance of the model under different data distributions is crucial to evaluate its true effect. In this study, we used a four-fold cross-validation to validate the model, that is, each fold reserved 25% of the data for testing. Figure 13 shows the mean and standard deviation of different performance indicators. The proposed method achieved a sensitivity of 71.5% ± 7.74%, a specificity of 89.73% ± 2.12%, an accuracy of 80.7% ± 3.96%, and an AUC of 83.8% ± 5.63%, and an F1 of 78.13% ± 7.18%. These results show that our method can maintain good performance under different data distributions. The large standard deviation is due to the small data set, large inter-subject variability, and certain differences in the data distribution of the training set and the test set under different data splits.

## CONCLUSION

Since the brain MRI images of MCI patients have small changes and blurry shapes, and MRI contains a large amount of redundant information, resulting in poor performance of existing methods, a convolutional neural network that fuses the attention mechanism and parallel dilated convolution is proposed. This network uses ResNet18 as the basic framework, introduces an attention gate mechanism, highlights relevant information of the lesion area in the image and suppresses irrelevant areas, eliminates redundant information in MRI images, and improves the ability to mine detailed information. At the same time, parallel dilated convolution is added to obtain a larger receptive field without downsampling, thereby enhancing the ability to acquire contextual information while maintaining detailed information on large-scale feature maps, thereby improving the classification performance of the network. Experimental results on the public dataset ADNI show that the model proposed in this study shows excellent performance in MCI

detection and effectively improves the detection rate of MCI screening. This discovery not only has important clinical significance for MCI patients themselves, but also provides strong support for doctors' clinical decision-making.

Although the method has significant progress in MCI detection, there is still potential for further improvements. The study mainly focused on structural MRI data, and future research could explore other types of image data, such as functional MRI, electroencephalography (EEG), and PET scans. These different types of data often provide different perspectives and information levels, and their comprehensive use is expected to improve the diagnostic accuracy and comprehensiveness of MCI. In addition, experimental verification on public data sets provides preliminary evidence of the effectiveness of this method, but in subsequent studies, the model needs to be further validated and applied in real clinical settings. In a real clinical environment, the model may face more challenges and complexities, such as differences between data collected by different devices, missing data, *etc.*, which all need to be taken into account.

### Funding
The work is supported by the Anhui Provincial Major Science and Technology Project (No. 202103a07020004, 202303a07020006-4), the Anhui Provincial Clinical Medical Research Transformation Project (No. 202204295107020004), the Hefei Fourth People's Hospital In-hospital Project (No. HFSY2020YB08) and the National Natural Science Foundation of China (No. 62133004). The funders had no role in study design, data collection and analysis, decision to publish, or preparation of the manuscript.

### Grant Disclosures
The following grant information was disclosed by the authors:
Anhui Provincial Major Science and Technology Project: No. 202103a07020004, 202303a07020006-4.
Anhui Provincial Clinical Medical Research Transformation Project: No. 202204295107020004.
Hefei Fourth People's Hospital In-hospital Project: No. HFSY2020YB08.
The National Natural Science Foundation of China: No. 62133004.

### Competing Interests
The authors declare there are no competing interests.

### Author Contributions
- Tao Wang conceived and designed the experiments, performed the experiments, analyzed the data, performed the computation work, prepared figures and/or tables, authored or reviewed drafts of the article, and approved the final draft.
- Zenghui Ding conceived and designed the experiments, performed the experiments, authored or reviewed drafts of the article, and approved the final draft.
- Xianjun Yang conceived and designed the experiments, authored or reviewed drafts of the article, and approved the final draft.

- Yanyan Chen conceived and designed the experiments, authored or reviewed drafts of the article, and approved the final draft.
- Yu Liu conceived and designed the experiments, performed the experiments, analyzed the data, authored or reviewed drafts of the article, and approved the final draft.
- Xiaoming Kong conceived and designed the experiments, performed the experiments, analyzed the data, authored or reviewed drafts of the article, and approved the final draft.
- Yining Sun conceived and designed the experiments, authored or reviewed drafts of the article, and approved the final draft.

## Data Availability

The MCI image dataset is available in the ADNI database (https://adni.loni.usc.edu/) and the code is available in the Supplementary File.

## Supplemental Information

Supplemental information for this article can be found online at http://dx.doi.org/10.7717/peerj-cs.2056#supplemental-information.

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
