# Peer review of "Detection of mild cognitive impairment based on attention mechanism and parallel dilated convolution"

_PeerJ Computer Science, doi:10.7717/peerj-cs.2056_

## Round 0.1 · original submission · Major Revisions

Please address all reviewers' comments.

**Language Note:** PeerJ staff have identified that the English language needs to be improved. When you prepare your next revision, please either (i) have a colleague who is proficient in English and familiar with the subject matter review your manuscript, or (ii) contact a professional editing service to review your manuscript. PeerJ can provide language editing services - you can contact us at [email protected] for pricing (be sure to provide your manuscript number and title). – PeerJ Staff

Reviewer 1 ·

Basic reporting

All comments have been added in detail to the 4th section called additional comments.

Experimental design

All comments have been added in detail to the 4th section called additional comments.

Validity of the findings

All comments have been added in detail to the 4th section called additional comments.

Additional comments

Review Report for PeerJ Computer Science
(Detection of mild cognitive impairment based on attention mechanism and parallel dilated convolution)

1. Within the scope of the study, a deep learning-based approach was proposed by performing various applications on the open source Alzheimer's Disease Neuroimaging Initiative dataset to detect mild cognitive impairment.

2. In order to clarify the place of the problem discussed in the Introduction section in the literature, it is recommended to add a literature table consisting of columns such as "author, methodology, dataset, evaluation metrics, performance, advantages, limitations".

3. The difference of the study from the literature and its main contributions to the literature should be stated more clearly and clearly at the end of the Introduction section, just before the Methods.

4. The dataset used in the study for classification was not obtained from a hospital with ethics committee permission specific to the study, but instead an open source dataset on which a lot of work has been done was used. For this reason, when examined in terms of the dataset, the originality point is very limited.

5. Dataset distribution was determined as 75% for training and 25% for testing. The results obtained in classification problems are very dependent on the dataset. For this reason, it should be clearly explained why the dataset distribution is preferred in this way, whether any different dataset distribution is made, and why cross-validation is not preferred.

6. It is very important to perform data preprocessing and augmentation operations rather than using the dataset raw. The change in the amount of data after augmentation should be expressed more clearly. The classification results should be compared before and after augmentation/preprocessing and the positive effect of these processes on the results should be expressed.

7. It has been stated that ResNet is used as the backbone network. When the literature is examined, there are many different deep learning-based networks that can be used as backbones. It should be explained why ResNet was preferred within the scope of the study and whether any other application was tried.

8. Within the scope of the study, a study-specific deep learning-based approach consisting of ResNet, Attention Gate Mechanism and Atrous Spatial Pyramid Pooling parts has been proposed. Although there are aspects of the approach that can be improved, the results obtained require more detailed analysis.

9. In order to analyze the classification results correctly, it is very important to obtain the evaluation metrics completely and accurately. When the study is examined from this perspective, there are many missing metrics and graphics. Missing metrics such as confusion matrix, ROC curve, validation-epoch graph need to be added.

10. The framework/toolbox information used in the application part needs to be added in more detail. In addition, it should be explained how the hyperparameters (optimizer, learning rate, etc.) used in the study were selected. Why was Adam optimizer preferred? Apart from this, have you tried different optimizers? How was the batch size? Why is the iteration limited to 50? Does increasing the iteration value have any positive/negative effect on the result? Why?

As a result, although the study is important in terms of the problem addressed, there are many aspects that can be improved. For this reason, the parts mentioned above must be tested and/or explained in detail.

Annotated reviews are not available for download in order to protect the identity of reviewers who chose to remain anonymous.

·

Basic reporting

Please change the sentence ‘and early diagnosis and intervention..’ in line 14 to ‘and an early diagnosis and intervention can delay its progression’.
Please replace the word ‘and’ with a comma between the words image and suppresses in line 19.
Please improve the grammar in lines 42 to line 44.
Text from line 88 to line 97 is repeated from the introduction.
Please provide a reference or name of the reports for the statements in line 36 to line 40.
Please provide reference for line 42 to line 44.
Please provide a reference for line 48 and 49.
Please provide a reference in line 63.
Please correct the reference in line 64.
Please use a standardised and correct style of citation either parenthetical or narrative throughout the text.
Centralize figure 1.

Experimental design

No Comments.

Validity of the findings

Please verify the increase in ACU by 6.85%? Line 317.
Please define the ‘overall performance’ of your model in line 333.
Please enrich your conclusion section for future researchers.

---

## Round 0.2 · accepted · Accept

Considering the comments of previous reviews, the authors addressed all comments accurately. The min or issues raised by Reviewer 2 were accurately addressed, and the same for the comments of Reviewer 1 that confirmed the corrections. Congratulations!

Reviewer 1 ·

Basic reporting

All comments have been added in detail to the 4th section called additional comments.

Experimental design

All comments have been added in detail to the 4th section called additional comments.

Validity of the findings

All comments have been added in detail to the 4th section called additional comments.

Additional comments

Review Report for PeerJ Computer Science
(Detection of mild cognitive impairment based on attention mechanism and parallel dilated convolution)

Thanks for the revision. I reviewed all the comments and the revised version in detail. The answers given and the changes made are appropriate. I recommend that this paper be accepted due to its contribution to the literature and its originality at a certain level. I wish the authors success in their new study. Best regards.